# Intention to donate blood and its predictors among adults of Gondar city: Using theory of planned behavior

**Ayenew Kassie**[ID]*, **Telake Azale, Adane Nigusie**

Department of Health Education and Behavioral Science, Institute of Public Health, College of Medicine and Health Sciences, University of Gondar, Gondar, Ethiopia

* kassieayenew@gmail.com

**Data Availability Statement:** The datasets generated and/or analyzed during the current study are available at University of Gondar, College of medicine and Health Science, Institute of Public Health and Gondar City health department in hard

## Abstract

### Background

Blood transfusion is a lifesaving procedure when someone encounters severe anemia, accident or injury, surgery, heavy bleeding during childbirth and cancer chemotherapy. The average blood donation rate of Africa is 4.7/1000 inhabitants and Ethiopia is among one of the countries with the lowest annual donation rate which is 0.8/1000 population. This study assessed intention to donate blood on adults of Gondar city administration using the theory of planned behavior.

### Methods

A community-based cross-sectional study was conducted. The study was conducted on two randomly selected Gondar sub-cities using systematic sampling on a sample size of 524 adults. Epi Data version 3.0 and STATA version 14 were used for entry and analysis of data respectively. Multiple linear regression was carried out to see the association between intention and sociodemographic variables, past donation experience, attitude, subjective norm and perceived behavioral control and with 95% confidence interval and a p-value of less than 0.05 was used to detect statistical significance.

### Results

A total of 515 respondents participated in the study giving a response rate of 98%. Most of the participants were females (66.4%) and the participants' age ranges from 18 to 65 years. The variance explained by the model was 49%. The mean intention to donate blood was 3.02±1.13. Direct perceived behavioural control (β = 0.14, CI (0.04, 0.23)), direct subjective norm (β = 0.11: CI (0.04, 0.17), direct attitude (β = 0.03; CI (0.01, 0.06)) and past behaviour of blood donation (β = 0.3; CI (0.07, 0.51) were significant predictor of intention.

### Conclusion

Theory of planned behavior could be successfully applied in determining adult's blood donation intention. Predictors of intention to donate blood were past experience of blood

and soft copy repository [www.UoG.edu.et]. In
addition the data are available from the authors
upon reasonable request and with permission of
the principal investigators (Ayenew Kassie- E-mail
kassieayenew@gmail.com).

**Funding:** The funders (University of Gondar) had
no role in study design, data collection and
analysis, decision to publish, or preparation of the
manuscript.

**Competing interests:** The authors have declared
that no competing interests exist.

**Abbreviations:** ATT, attitude; CI, confidence
interval; PBC, perceived behavioral control; SN,
subjective norm; TPB, theory of planned behavior;
VIF, variance inflation factor.

donation, direct subjective norm, direct perceived behavioural control and direct attitude.
None of the external variables predict blood donation intention.

## Introduction

Blood is essential to life and circulates through our body and delivers oxygen and nutrients to the body's cells. It has no substitute and cannot be made or manufactured. Generous blood donors are the only source of blood for patients in need of a blood transfusion [1–3].

Blood transfusion is a lifesaving procedure given to someone in need of it when encounters anemia(severe iron deficiency, thalassemia or aplastic anemia), accident or injury, undergo major surgery, during childbirth and cancer chemotherapy[4].

Globally millions of lives have been saved through blood transfusion each year. However, its availability is the main concern of developing countries[5]. The average blood donation rate of Africa is 4.7/1000 inhabitants and Ethiopia is among one of the countries with the lowest annual donation rate (0.8/1000) population [6].

According to the Ethiopia Health sector transformation plan (HSTP) report of 2015/16, it was planned to collect 202,000 units of blood but the collected was 169, 744 units of blood [7].

Providing adequate and safe blood to those who need blood is the responsibility of blood banks. Blood could be obtained from voluntary non- remunerated donors, family replacement remunerated or paid donors, and autonomous donors[8] but the safest way is blood obtained from voluntary non- remunerated donors.

According to the theory of planned behavior, human action is guided by three kinds of considerations. The first is behavioral beliefs which is beliefs about the likely outcomes of the behavior and the evaluations of these outcomes. The second is normative beliefs; it is beliefs about the normative expectations of others and motivation to comply with these expectations. The last is control beliefs which is beliefs about the presence of factors that may facilitate or impede performance of the behavior and perceived power of these factors[9]. Intention is the most proximal and best predictor of behaviour and intention is further predicted by the attitude towards blood donation[10], subjective norms associated with blood donation and perceived behavioral control over blood donation. Across different researches theory of planned behavior has a potential to predict blood donation, and the predictors of theory planned behavior explained between 51% and 80.7% of variances to donate blood[9–11]. Therefore this study aims to assess intention towards blood donation of adults using the theory of planned behavior[12].

## Rationale of the study

There is a deficit between the amount of blood needed and collected. Different studies conducted in Ethiopia on blood donation show that the rate of blood donation is still very low. In developing countries like Ethiopia the most source of blood is obtained from voluntary individuals, and most of it is collected through campaign because there are no regular blood donors. This study assessed predictors (attitude, subjective norm, perceived behavioural control and past donation behaviour) of intention to donate blood. Furthermore, the study finding may help policy makers and health professionals in developing new approach for increasing blood donation practice among adults.

## Methods

### Study design and setting

A community-based cross-sectional study was conducted from 1[st] March to 30[th] March 2019 among adults of Gondar city to assess intention to donate blood and its predictors using theory of planned behavior. Gondar city is located about 727 km away from Addis Ababa, the capital city of Ethiopia, 180 km away from Bahir Dar the capital city of Amhara Regional State. Gondar city has a total area of 192.3Sq.KM.

### Study population and sampling

All Adults whose ages between 18 and 65 years old and who are residents of Gondar city were included. Those Critically ill Individuals who couldn't give information were excluded.

The sample size of the study was calculated using a single population for intention to donate blood and a double population means for predictors. The final sample size of the study was 524 adults.

From a total of six sub-cities of Gondar city, two sub-cities were randomly selected using a simple random sampling technique. Then a systematic sampling technique was employed to select households from each sub-city. The interval value (K) was calculated for selected sub-cities by dividing the total number of households of each selected sub-city to the proportional sample size of the sub-city. The initial household to be interviewed was selected randomly with a lottery method.

### Data collection procedure

Data were collected with an interviewer-administered pretested structured questionnaire prepared by the investigator after reviewing different relevant literatures[9, 11–15]. The questionnaire was consists independent variables such as socio-demographic characteristics, knowledge questions, past experience, and theory of planned behavior variables (direct and indirect attitude, direct and indirect subjective norm, direct and indirect perceived behavioral control) and the dependent variable intention.

### Measurements

**Intention.** Intention to donate blood was measured by using three items[12]. "I decided to donate blood in the next three months, "I want to donate blood in the next three months and "I would like to donate blood in the next three months". Responses ranged "strongly disagree" (1) to "strongly agree" (5). A composite score was summed up by all the items(Table 1). The expected minimum and maximum score was one and five respectively. The internal consistency of the items were (Cronbach's α = 0.96).

**Attitude.** Direct attitude towards blood donation was assessed by five items." for me donating blood in the next three months is pleasant/unpleasant, bad/good, satisfying/unsatisfying, useful/harmful. . .. . .. "On five point Likert scale. A composite score of direct attitude was obtained by summing up all the five items. The expected minimum and maximum score was five and twenty five respectively. The internal consistency of direct attitude was (Cronbach's α = 0.83). Five items were used to measure behavioral belief with responses ranged from "strongly disagree agree"(1) to "strongly agree"(5). Outcome evaluation of blood donation beliefs were measured by asking participants to evaluate the five salient beliefs consequences of blood donation. Each behavioral belief was multiplied by the outcome evaluation to produce a new variable an indirect attitude. A composite score of an indirect attitude was obtained by summing up all the five products of behavioral belief and outcome evaluation.

**Table 1. Summary of direct and indirect measures of theory of planned behavior variables.**

|  | Item | Scoring | Outcome |
|---|---|---|---|
| Direct measure |  |  |  |
| I. Attitude(ATT) | 5 | $\sum_{i=1}^{5}(ATTi)$ | Direct ATT score |
| II. Subjective norm(SN) | 5 | $\sum_{i=1}^{5}(SNi)$ | Direct SN score |
| III. Perceived behavioral control(PBC) | 5 | $\sum_{i=1}^{5}(PBCi)$ | Direct PBC score |
| IV. Intention(I) | 3 | $\sum_{i=1}^{3}(Ii)$ | Intention score |
| Indirect measures |  |  |  |
| I. Behavioral belief(BB) | 5 | $\sum_{i=1}^{5}(BB*OE)$ | Indirect ATT score |
| II. Outcome evaluation(OE) | 5 |  |  |
| III. Normative belief(NB) | 4 | $\sum_{i=1}^{4}(NB*MC)$ | Indirect SN score |
| IV. Motivation to comply(MC) | 4 |  |  |
| V. Control belief(CB) | 5 | $\sum_{i=1}^{5}(CB*PC)$ | Indirect PBC score |
| VI. Power of control(PC) | 5 |  |  |
| VII. Knowledge(Kng) | 10 | $\sum_{i=0}^{1}(Kngi)$ | Knowledge score |

**Subjective norm.** A total of five items were used to assess the direct subjective norm. The score ranged from 1 to 5 and had high internal consistency (Cronbach's α = 0.77). A composite score of the direct subjective norm was obtained by summing up all the five items. The expected minimum and maximum score was five and twenty five respectively. A total of four items were used to assess normative belief and the response ranged from 1 to 5. Each normative belief statement converted into four corresponding motivations to comply with items. Each normative belief was multiplied by the motivation to comply to produce indirect subjective norm. A composite score of the indirect subjective norm was obtained by summing up all the four products of normative belief and motivation to comply.

**Perceived behavioral control.** Direct perceived behavioral control was assessed by five items. All of them were Likert scale questions. The score ranged from 1 to 5 and had high internal consistency (Cronbach's α = 0.78). A composite score of direct perceived behavioral control was obtained by summing up all the five items(Table 1). The expected minimum and maximum score was five and twenty five respectively. Five items were used to measure control belief with responses ranged from 1 to 5. Each control belief statement converted into five corresponding power of control items. Each control belief was multiplied by the power of control to produce an indirect perceived behavioral control. A composite score of indirect perceived behavioral control was obtained by summing up all the five products of control belief and power of control.

**Knowledge.** Knowledge towards blood donation was assessed using ten questions. The questins were developed from reviewing different literatures which had similar demographic characteristics[16–18]. Some of the items were measured as yes or no and others were measured with multiple response options. Some of the items had response option of "don't know". For example the question "What is the appropriate age to donate blood?" had response options of "<18, 18–65, >65 and don't know". Each response were scored with value of "1" for correct response and for incorrect response given value of "0". Individuals who respond "don't know"

were considered as incorrect. Knowledge score of each individuals was obtained by summing up all the ten items and the expected score ranged from 0 to 10.

## Data processing and analysis

All collected data were entered into EpiData version 3.0 and exported to STATA version14 statistical software for its analysis. Descriptive analysis was used to see frequency distribution, mean and standard deviation. Correlation analysis was done between indirect and direct theory of planned behavior (TPB) variables to see the correlation between them. Multiple Linear regression analysis was computed to test the strength and direction of association between the dependent variable and independent variables. $R^2$ was used for the ability of explanatory variables to explain dependent variables. An unstandardized β coefficient was used to interpret the effect of predictors on the intention to donate blood. The assumption of normality was checked statistically and it was normally distributed. Test of homoscedasticity using white's test was conducted. All results supported the assumption of homoscedasticity. Linearity assumption was checked using a scatter plot of the standardized residuals versus the predicted values from the regression analysis. Multicollinearity assumptions were tested by the variance inflation factor (VIF) and the value of all variables was below ten. The assumption of an outlier was tested using Cook's and there was no outlier[19]. Variables with a p-value of less than 0.05 at 95%confidence intervals were considered as statistically significant.

## Ethical consideration

Ethical clearance was obtained from the Institutional Review Committee of the institution of Public Health, College of Medicine and Health Sciences, University of Gondar. A letter of permission was obtained from Gondar city administrative health office. After the purpose and objective of the study have been informed, verbal consent obtained from each study participant. All participants were informed that to participate voluntarily and they can withdraw from the study at any time if they were not comfortable about the questionnaire. To keep confidentiality of any information provided by study subjects, the data collection procedure was anonymous.

# Results

## Socio-demographic characteristics

A total of 515 respondents have participated with a response rate of 98%. Most of them (66.4%) were females and the participants' age were range from 18 to 65 years. The mean age of participants with standard deviation was (32.25 years ±9.32 years). Nearly 88.3% were orthodox Christian in religion and 80.58 were Amhara in ethnicity (Table 2).

## Knowledge about blood donation

Respondents were asked ten questions that assess knowledge of blood donation. For each knowledge item scores were summed up to get over all knowledge score, individuals correctly answered the item given value of "1" and for those answered incorrectly valued"0" and then mean and standard deviation were calculated. Accordingly, the mean and standard deviation of knowledge score was 6.23±2.07. More than half (55.53%) of participants correctly knew the appropriate age category of blood donation (18–65 years old), with respect to the amount of blood donated at one time 137(26.6%) knew that it is transfused 350ml to 450 ml and 150 (29.13%) knew the minimum weight required to donate blood is 45 kilograms (Table 3).

**Table 2. Socio-demographic characteristics of Gondar city adults, North West Ethiopia, 2019(N = 515).**

| Variable | Category | Frequency | Percent |
|---|---|---|---|
| Age | 18–30 | 264 | 51.3 |
| | 31–40 | 156 | 30.3 |
| | 41–50 | 72 | 13.0 |
| | > = 51 | 23 | 4.5 |
| Marital status | Single | 219 | 42.6 |
| | Married | 215 | 41.8 |
| | Divorced | 62 | 11.4 |
| | Widow | 19 | 3.7 |
| Educational status | Unable to read and write | 60 | 11.7 |
| | Able to read and write | 78 | 15.2 |
| | Grade1-8 | 35 | 6.8 |
| | Grade 9–12 | 114 | 22.1 |
| | Diploma | 123 | 23.9 |
| | Degree and above | 105 | 20.4 |
| Ethnicity | Amhara | 415 | 80.6 |
| | Kimant | 88 | 17.1 |
| | Others | 12 | 2.3 |
| Religion | Orthodox Christian | 455 | 88.4 |
| | Muslim | 51 | 9.9 |
| | Others | 9 | 1.8 |
| Occupation | Government employee | 157 | 30.5 |
| | Private employee | 122 | 23.7 |
| | Merchant | 52 | 10.1 |
| | Student | 42 | 8.2 |
| | No job | 83 | 16.1 |
| | Others | 59 | 11.5 |
| Income(Birr /month) | <500 | 103 | 20.0 |
| | 500–1000 | 113 | 22.0 |
| | 1001–1500 | 40 | 7.8 |
| | > = 1500 | 259 | 50.3 |

**Past experience of blood donation.** Of the total respondents, 80(15.53%) had ever practiced at least once donate blood in their lifetime. From those 38(47.50%) were donated once, 33(41.25%) had donated twice, 6(7.50%) thrice and 3(3.75%) had donated more than three times. Individuals who didn't donate blood mention their reasons of those 228(52.41%) had no information when and where blood donated and 42(9.66%) of them mentioned other different reasons such as weight is low than required, not interested to give their blood to other than family. Furthermore, Respondents were asked whether they or their families required a blood and 29(5.63%) of them had required, of those 22(75.86%) had been transfused.

**Theory of planned behaviour variables.** The mean score of the indirect subjective norm was 50.94±17.82 and mean score of the indirect perceived behavioral control was 49.52± 24.44 (Table 4).

**Magnitude of intention.** The magnitude of intention was assessed using three items of the five-point Likert scale. The mean intention of respondents was 3.02 with a standard deviation of 1.13. The reliability test of Cronbach's alpha was (ά = 0.96).

**Table 3. Knowledge of adults on blood donation at Gondar city, North West Ethiopia, 2019(N = 515).**

| Variable | Category | Correct answer | Incorrect answer |
|---|---|---|---|
| Highly bleed individuals with an accident needs blood | *Yes* | 482(93.6%) | 33(6.4%) |
| | No | | |
| Mothers bleed during delivery need blood | *Yes* | 483(93.8%) | 32(6.2%) |
| | No | | |
| Patients undergo major surgery need blood | *Yes* | 406(79.0%) | 109(21.2%) |
| | No | | |
| Severe anemic patients need blood | *Yes* | 417(81.0%) | 98(19.0%) |
| | No | | |
| What is the appropriate age to donate blood | <18 | 286(55.6%) | 229(44.5%) |
| | *18–65* | | |
| | >65 | | |
| | Don't know | | |
| Who are eligible to donate blood | Males only | 441(85.7%) | 74(14.4%) |
| | Females only | | |
| | *Both* | | |
| Amount of blood donated at one time | *350-450ml* | 137(26.6%) | 378(73.4%) |
| | 500ml | | |
| | 200ml | | |
| | Don't know | | |
| Duration of blood on the shelf | 1 month | 128(24.9%) | 387(75.1%) |
| | 2 months | | |
| | *3 months* | | |
| | >3 months | | |
| | Don't know | | |
| Interval of blood donation | Every week | 279(54.2%) | 236(45.8%) |
| | Every month | | |
| | *Every 3 months* | | |
| | Every 6 months | | |
| | Every year | | |
| | Don't know | | |
| Minimum weight for blood donation | < 45 kg | 150(29.1%) | 365(70.9%) |
| | *45kg* | | |
| | 50kg | | |
| | 55kg | | |
| | Don't know | | |

words/phrases which are written italic are correct answers

**Correlation of theory of planned behavior variables and other variables with intention.** All predictors of the intention of theory of planned behavior variables had positively and significantly correlated with intention at a p-value of <0.05. Direct subjective norm had the highest correlation followed by direct perceived behavioral control, direct attitude, past experience of blood donation and indirect subjective norm. Indirect perceived control had the least correlation. All indirect measures positively correlated with their direct measures. Indirect attitudes with direct attitude and indirect subjective norm with direct subjective norm had medium correlation whereas indirect perceived behavioral control with direct perceived behavioral control had a weak correlation[20] (Table 5).

**Table 4. Distribution of TPB variables on intention to donate blood of Gondar city adults, North West Ethiopia, 2019(N = 515).**

| Variable | Mean | Standard deviation | Minimum | Maximum |
|---|---|---|---|---|
| Direct attitude | 18.1 | 3.5 | 8 | 25 |
| Direct subjective norm | 17.2 | 3.6 | 6 | 25 |
| Direct PBC | 17.4 | 3.4 | 8 | 25 |
| Indirect attitude | 65.2 | 34.6 | 8 | 125 |
| Indirect subjective norm | 50.9 | 17.8 | 7 | 100 |
| Indirect PBC | 49.5 | 24.4 | 8 | 125 |
| Intention | 3.0 | 1.1 | 3 | 15 |

**Simple linear regression.** Prior to the analysis the assumptions of linear regression were checked as mention above in method part. Then Simple linear regression was performed to assess the association of each independent variable with intention to donate blood at 95% confidence interval. Variables which were significant in simple linear regression were entered to multiple linear regression for further statistical significance. Sociodemographic variables (sex, educational status, marital status, income and occupation), knowledge, past experience of blood donation and all direct measures of TPB variables were candidates for multiple linear regression model.

**Multiple linear regression.** First TPB variables such as direct attitude, direct subjective norm and direct perceived behavioral control were entered to the regression. These variables explain the model 36.5%. Then past experience of blood donation were added to TPB variables and explain the model 40.4%. Finally other external variables (sociodemographic variables and knowledge) were added.

The variance explained by the intention to donate blood from all predictors was 49%. In multiple linear regression variables which found statistically significant at 5% level of significant were direct subjective norm ($\beta$ = 0.11; CI (0.04, 0.17), direct perceived behavioural control ($\beta$ = 0.14, CI (0.04, 0.23)), Direct Attitude ($\beta$ = 0.03; CI (0.01, 0.06) and past behaviour of blood donation ($\beta$ = 0.3; CI (0.07, 0.51). The standardized regression coefficient suggested that direct subjective norm is by far the most important predictor of intention to donate blood followed by direct perceived behavioral control, direct attitude and past experience of blood donation. The model result showed that previous experience of blood donation affects significantly and positively intention to blood donation. Respondents who had previous experience of blood donation were 0.3 times more intended than those who didn't have experience provided that other variables are kept constant. Direct subjective norm positively and significantly

**Table 5. Pearson correlation of TPB variables among adults of Gondar city, North West Ethiopia, 2019(N = 515).**

| Variables | 1 | 2 | 3 | 4 | 5 | 6 | 7 | 8 |
|---|---|---|---|---|---|---|---|---|
| 1. Intention | | | | | | | | |
| 2. Direct ATT | 0.4* | | | | | | | |
| 3. Direct SN | 0.53* | 0.7* | | | | | | |
| 4. Direct PBC | 0.46* | 0.6* | 0.61* | | | | | |
| 5.Indirect ATT | 0.48* | 0.65* | 0.57* | 0.54* | | | | |
| 6.Indirect SN | 0.27* | 0.37* | 0.47* | 0.44* | 0.41* | | | |
| 7.Indirect PBC | 0.14* | 0.11* | 0.12* | 0.25* | 0.07 | 0.47* | | |
| 8. knowledge | 0.30* | 0.38* | 0.35* | 0.28* | 0.19* | 0.31* | 0.26* | |

* = Correlation is significant at 0.05(2 tailed)

**Table 6. Multiple linear regression of intention to donate blood and its predictors among adults of Gondar city, North West Ethiopia, 2019(N = 515).**

| Variable | | Unstandardized B | Standardized β | 95% CI for B |
|---|---|---|---|---|
| Constant | | -4.61 | | -12.04,2.82 |
| Sex | Female(ref.) | | | |
| | Male | 0.01 | 0.01 | -15, 15 |
| Marital status | Single(ref) | | | |
| | Married | 0.02 | 0.01 | - 0.15, 0.20 |
| | Divorced | 0.11 | -0.01 | -0.19,0.41 |
| | Widowed | 0.09 | -0.01 | -0.36, 0.55 |
| Occupation | No job(ref.) | | | |
| | Gov't employee | 0.20 | -0.04 | -0.08, 0.48 |
| | Private employee | 0.06 | -0.05 | -0.18, 0.31 |
| | Merchant | 0.31 | -0.01 | -0.01, 0.63 |
| | Student | 0.23 | 0.01 | -0.08, 1.19 |
| Income in ETB | <500(refer) | | | |
| | 500–1000 | 0.02 | 0.02 | -0.76, 0.81 |
| | 1001–1500 | -0.78 | -0.01 | -1.85, 0.28 |
| | >1500 | -0.39 | 0.01 | -1.19, 0.41 |
| Knowledge | | 0.48 | 0.03 | -0.09, 0.18 |
| Experience of blood donation | No(ref) | | | |
| | Yes | 0.30* | 0.08 | 0.07, 0.51 |
| Direct attitude | | 0.03* | 0.10 | 0.01, 0.06 |
| Direct SN | | 0.11* | 0.50 | 0.04, 0.17 |
| Direct PBC | | 0.14* | 0.14 | 0.04, 0.23 |

*p<0.05, ref. = reference

influence the intention to donate blood. For a score increase of direct subjective norm intention to donate blood increase by score of 0.11 provided that other variables are kept constant. Keeping other variables constant, for a positive score increase of direct perceived behavioral control intention to donate blood increase by score of 0.14. A positive score increase in attitude could result in an increase of intention to donate blood by score of 0.03 provided that other variables kept constant (Table 6).

Here in Fig 1 shows that All of the indirect measures significantly associate with their direct measures and further those direct measures (attitude, subjective norm and perceived behavioural control) had significantly predict blood donation intention. All indirect measures of TBP variables had significant correlation each other. Direct subjective best explained intention ($R^2$ = 34), direct perceived behavioural control explained 27% ($R^2$ = 0.27) and direct attitude explained 25% ($R^2$ = 0.25). Those studies might be familiar with Likert scale questions so that they could easily understand.

## Discussion

In this study, the intention of blood donation and its predictors among adults were assessed using the theory of planned behavior. Subjective norm, Attitude, perceived behavioral control from the theory of planned behavior variables and past experience of blood donation were predictors of intention to donate blood in the next three months. The model explained 49% of the variance in intention to donate blood. This finding is lower than the study conducted in Ireland [9] and Pakistan [21]. This difference may be due to socio-economic and socio-cultural

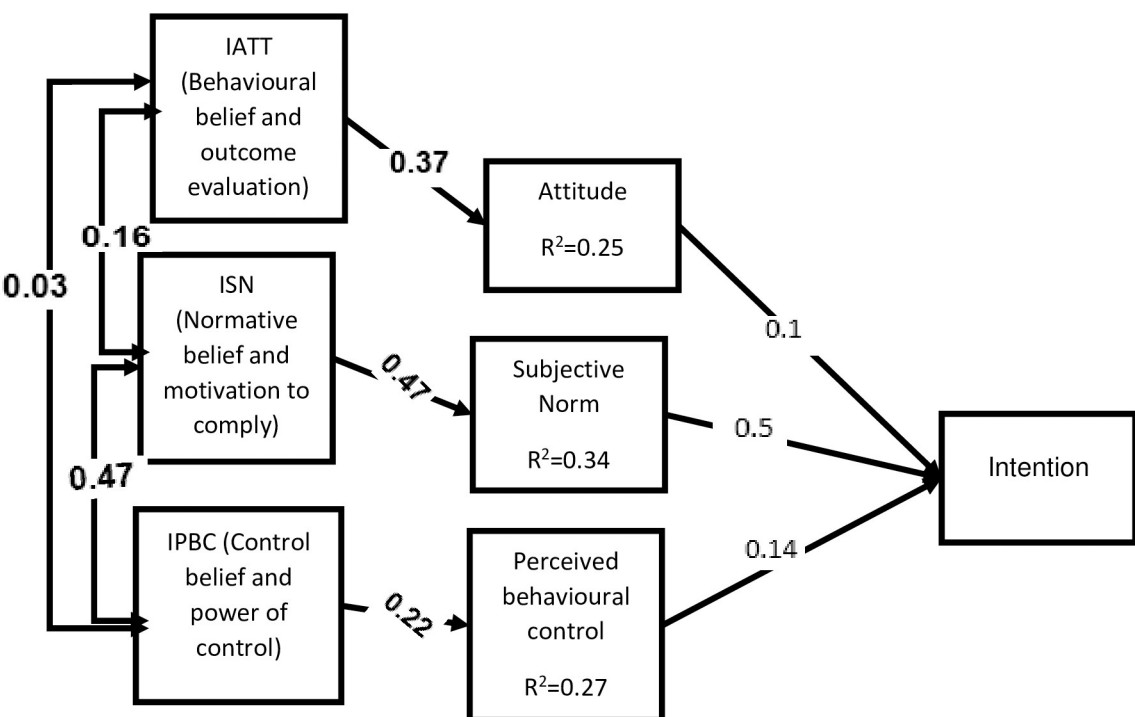

**Fig 1. Standardized path coefficient of theory of planned behavior variables (all significant at p<0.05).** IATT = indirect Attitude; ISN = indirect Subjective norm; IPBC = indirect perceived behavioral control.

variation, respondents in those studies might be familiar with Likert scale questions so that they could easily understand it [22].

The present study revealed that external variables such as all socio demographic variables and knowledge had no significant effect on intention to donate blood in the next three months. This means prediction of intention to donate blood don't vary among individuals having different sociodemographic characteristics and knowledge level towards blood donation.

In this study, the mean intention to donate blood in the next three months was neutral. This finding is consistent with a study conducted in Mekelle, Dire Dawa and Israel[10, 23, 24].

The result of our finding revealed that direct subjective norm is a predictor of intention to donate blood. This is comparable with studies conducted in China and in Ireland[9, 25]. This suggests that decisions made in this context do not only concern the respondents but also families, friends, relatives, and health professionals. Thus, interventions to improve blood donation practice should also target those important others (families, friends, relatives, and health professionals) as a whole rather than focusing only on the individual who are eligible to donate blood donation. If their important others are engaged in blood donation practice, eligible individuals are more likely to participate in this practice.

The current study also reported that direct perceived behavioral control is the predictor of intention to donate blood in the next three months. This finding is supported by a study conducted in Ireland [9] and a study in Pakistan [21]. This is because those individuals who perceived that donation blood in the following three months is easy and those individuals who are confident enough to donate were more likely to donate blood. This suggests that encouraging individuals to aim for donating blood should involve consideration of factors that is under their control. That means those individuals who able to control the fear of pain from the needle, control fear of fainting after donation, control fear of becoming anemic following giving

blood and control fear of being susceptible to certain health problems after donation so that they may lead to increased intention to donate blood. These results support Ajzen's theoretical assumptions; the more individuals have a high degree of control over factors that facilitate or prevent them to donate blood; the greater will be their intention to donate blood[12].

This study prevailed that direct attitude was a predictor of intention to donate blood in the next three months. This is consistent with studies conducted in China, Pakistan and Botswana [21, 25, 26]. This implies that health education interventions should be given which targeted on creating favourable attitude towards blood donation. Those individuals who believed that blood donation could reduce death of mothers, save life of anemic patients and gives internal satisfaction were more likely to had favourable attitude and then more intended to donate blood.

## Conclusion

This study revealed that past experience of blood donation, direct subjective norm, direct perceived behavioral control, and direct Attitude predict the intention of blood donation in the next three months. All indirect measures were significantly and positively correlated with their respective direct measures. None the external variables such as socio-demographic factors and knowledge were significant predictors of intention to donate blood.

## Acknowledgments

We would like to forward our heartfelt gratitude to the University of Gondar College of Medicine and Health Sciences Institute of Public Health for providing us ethical clearance and assigning adivisorship to do this practical research.

Finally, we would like to acknowledge study participants, data collectors and supervisors for their time and contribution to this work.

## Author Contributions

**Conceptualization:** Ayenew Kassie.

**Data curation:** Ayenew Kassie.

**Formal analysis:** Ayenew Kassie.

**Funding acquisition:** Ayenew Kassie.

**Investigation:** Ayenew Kassie.

**Methodology:** Ayenew Kassie, Telake Azale, Adane Nigusie.

**Software:** Ayenew Kassie.

**Supervision:** Ayenew Kassie, Telake Azale, Adane Nigusie.

**Validation:** Ayenew Kassie, Telake Azale, Adane Nigusie.

**Writing – original draft:** Ayenew Kassie.

**Writing – review & editing:** Ayenew Kassie, Telake Azale, Adane Nigusie.

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
