## [Decision Letter · Decision Letter 0]

3 Oct 2019

PONE-D-19-22717

INTENTION TO DONATE BLOOD AND ITS PREDICTORS AMONG ADULTS OF GONDAR CITY: USING THEORY OF PLANNED BEHAVIOR

PLOS ONE

Dear Mr Kassie,

Thank you for submitting your manuscript to PLOS ONE. After careful consideration, we feel that it has merit but does not fully meet PLOS ONE’s publication criteria as it currently stands. Therefore, we invite you to submit a revised version of the manuscript that addresses the points raised during the review process.

As our academic advisors (reviewers) suggested, please consider to perform substantial amendments in the paper. You will see in their reviews how different methodological issues need attention and the data analysis (included its description and pertinence) can be substantially improved if more advanced techniques are eventually considered and pertinently explained.

Please remember that, according to the journal policies, it is mandatory to address all the comments received, by means a rebuttal letter, in which responses, rationales and detail on the modifications made in regard to each comment should be included.

We would appreciate receiving your revised manuscript by Nov 17 2019 11:59PM. To enhance the reproducibility of your results, we recommend that if applicable you deposit your laboratory protocols in protocols.io, where a protocol can be assigned its own identifier (DOI) such that it can be cited independently in the future. For instructions see: http://journals.plos.org/plosone/s/submission-guidelines#loc-laboratory-protocols

We look forward to receiving your revised manuscript.

Kind regards,

Sergio A. Useche, Ph.D.

Academic Editor

PLOS ONE

Journal Requirements:

1. Please include additional information regarding the survey or questionnaire used in the study and ensure that you have provided sufficient details that others could replicate the analyses. For instance, if you developed a questionnaire as part of this study and it is not under a copyright more restrictive than CC-BY, please include a copy, in both the original language and English, as Supporting Information.  If the original language is written in non-Latin characters, for example Amharic, Chinese, or Korean, please use a file format that ensures these characters are visible.

2. Please state whether you validated the questionnaire prior to testing on study participants. Please provide details regarding the validation group within the methods section.

Reviewers' comments:

Reviewer's Responses to Questions

**Comments to the Author**

1. Is the manuscript technically sound, and do the data support the conclusions?

Reviewer #1: Partly

Reviewer #2: Yes

Reviewer #3: No

2. Has the statistical analysis been performed appropriately and rigorously? 

Reviewer #1: Yes

Reviewer #2: Yes

Reviewer #3: No

3. Have the authors made all data underlying the findings in their manuscript fully available?

Reviewer #1: Yes

Reviewer #2: Yes

Reviewer #3: No

4. Is the manuscript presented in an intelligible fashion and written in standard English?

Reviewer #1: Yes

Reviewer #2: Yes

Reviewer #3: No

5. Review Comments to the Author

Reviewer #1: The paper looks at understanding the motivations of people to donate blood in two Gondar sub cities, this is fine, but the same does not necessarily seem to reflect the demographic population of Ethiopia and this at least needs to be mentioned.

I am not sure why the authors use regression analysis as this does not allow more complex relationships to be consider and also assumes each variable varies independent of others. Thus Structural equation modelling (which is often used when undertaking research using TPB) would strengthen the analysis.

The abstract also mentions that the mean score of giving is lower, but this does not seem to be the focus as the question is how do these other TPB variables influence this intention. The conclusion in the abstract needs to be more than state the results.

The intro seems to suggest that donation is the only recruitment method and in some African countries people are paid and thus it is not a generous behaviour.

I found the introduction and other choppy with multiple single sentence paragraphs. These are all fine, but the authors might craft a story around the issues.

For the measures I do think they need to identify the specific scales that were used. These are not referenced and I am not sure who measures were used. This is needed. There is no description of the test of the knowledge measure and how did they make sure their measure was appropriate? For the first question on this it reads as if people could select multiple answers as the sum is greater than the sample number? What did they do with the don’t know respondents are these correct or incorrect?

It is unclear if they ran a factor analysis to assess the measures or just Cronbach’s alphas? Some constructs don’t mention the alphas.

The demographics are fine, but they need to comment if this is reflective of the population.

Table 4 does not include knowledge?

The regression included demographics, which is fine, but thy cold explain this and why it was done. I am not sure why they did not use knowledge as a continuous variable? They also mentioned that they include past donation behaviour and this does not seem to be included and should.

I personally prefer more complex models rather than regression or at least some assessment of interaction effects as it is assumed the variables do not impact on each other.

The discussion discuses a range of factors such as fear of needles and I am not sure where this was included in the survey. I do think that there should be a brief discussion of the blood donation process in Ethiopia as it could also be that it I shard to access blood donation locations, which could then explain these results. I just don’t know if it is an issue in the discussion as it is presented.

The statement “Most individual’s intention to donate blood in the next three months were below the mean intention.” Needs to be rewritten as I think they mean it was below the scale midpoint. Can most people be below average?

Reviewer #2: Methods part, may need more explanation how several variables were measured. Furthermore, discussion part there is a need to focus of discussing the findings rather than repeating the result which are already interpreted on result section

Reviewer #3: The manuscript need English editing. The paper needs modification for methodology and results. Theory of planned behavior should be described in detail. The model used should also be explained in methods. Please see attached comments for revision.

6. PLOS authors have the option to publish the peer review history of their article (what does this mean?). If published, this will include your full peer review and any attached files.

Reviewer #1: No

Reviewer #2: No

Reviewer #3: Yes: Bushra Moiz

---

## [Author Response · Author response to Decision Letter 0]

6 Nov 2019

we have attached the response to each reviews.

---

## [Decision Letter · Decision Letter 1]

26 Nov 2019

PONE-D-19-22717R1

INTENTION TO DONATE BLOOD AND ITS PREDICTORS AMONG ADULTS OF GONDAR CITY: USING THEORY OF PLANNED BEHAVIOR

PLOS ONE

Dear Mr Kassie,

Thank you for submitting your manuscript to PLOS ONE. After careful consideration, we feel that it has merit but does not fully meet PLOS ONE’s publication criteria as it currently stands. Therefore, we invite you to submit a revised version of the manuscript that addresses the points raised during the review process.

The manuscript has been considerably improved for this version. Both reviewers agree on its value and pertinence, but some further changes are required. You will also find a sanitized copy attached to the comments, containing some additional queries and suggestions.

Please put especial attention to those comments related to the discussion and interpretation of the findings (also, all the conclusions should be properly supported by the data), and their practical implications. Also, be careful with the technical issues referred by our reviewers.

As a particular comment, I would ask the authors to de-capitalize the title of the manuscript.

We would appreciate receiving your revised manuscript by Jan 10 2020 11:59PM. To enhance the reproducibility of your results, we recommend that if applicable you deposit your laboratory protocols in protocols.io, where a protocol can be assigned its own identifier (DOI) such that it can be cited independently in the future. For instructions see: http://journals.plos.org/plosone/s/submission-guidelines#loc-laboratory-protocols

We look forward to receiving your revised manuscript.

Kind regards,

Sergio A. Useche, Ph.D.

Academic Editor

PLOS ONE

Reviewers' comments:

Reviewer's Responses to Questions

**Comments to the Author**

1. If the authors have adequately addressed your comments raised in a previous round of review and you feel that this manuscript is now acceptable for publication, you may indicate that here to bypass the “Comments to the Author” section, enter your conflict of interest statement in the “Confidential to Editor” section, and submit your "Accept" recommendation.

Reviewer #1: (No Response)

Reviewer #2: (No Response)

2. Is the manuscript technically sound, and do the data support the conclusions?

Reviewer #1: Partly

Reviewer #2: Partly

3. Has the statistical analysis been performed appropriately and rigorously? 

Reviewer #1: I Don't Know

Reviewer #2: Yes

4. Have the authors made all data underlying the findings in their manuscript fully available?

Reviewer #1: Yes

Reviewer #2: Yes

5. Is the manuscript presented in an intelligible fashion and written in standard English?

Reviewer #1: No

Reviewer #2: No

6. Review Comments to the Author

Reviewer #1: There are still some expression issues in the new sections. I cannot identify all of these but the paper needs to be professional copy edited.

For example:

“The amount of unit of blood needed by blood banks and actual collected unit of blood is not compatible. Should be that there is a deficit between the amount of blood needed and collected.

“In developing countries like Ethiopia the most source of blood is obtained from generous individuals, and most of it is collected through campaign because there is no trend of people going and providing blood in blood banks.” I am not sure what this is saying?

Therefore this study assessed the factors within the theory of planned behaviour which determine influence individuals to donate blood; including like their attitude, influence of other individuals (subjective norm), presence of facilitating and hindering factors, individuals level of perceived behavioural control and their previous experience of donation to future.

The elicitation study was conducted, to explore relevant salient behavioral beliefs, normative belief and perceived behavioural control.

five and twenty five respectively. (Other times they us 5 to 25)

Knowledge: knowledge towards blood donation was assessed using ten questions which were previously used items in similar research which had similar demographic characteristics.

The test for outlier was assessed using Cook's and there was no outlier. (Is this a method or a measure and should be clearer also do they need a reference?)

The authors need to indicate what do not know means for the knowledge scale, in the manuscript, not just in the response document (this is also not discussed in the knowledge section and there may be no don’t know category). Knowledge is also not described in Table 1. I am not sure you need to explain response for individual questions, as this appears in Table 3.

Correlation does not look at the relationship (i.e. inferring causality). Thus this might need to be rephrased.

In the paper it needs to explain that the sample is representative of the population. I do think this is odd given the composition (i.e. the population is not 66% female and a very high orthodox Christian sample). In the mention of the religion, it should read Christian orthodox means as there are many religions that refer to orthodox?

I want to thank the authors for the additions of the models, but I am not sure how these relate to multiple regression in Table 6? If this is a SEM model, then it needs to be better explained and discussed. If they don’t do this then they should cut the diagram.

The section labelled strength and limitation is an odd title?

Reviewer #2: Some of the comments were not addressed adequately, especially on results and discussion.There is need for the author to clearly and correctly interpret the results. Discussion part, he/she may need to familiarise oneself with the best way of discussing the findings .

7. PLOS authors have the option to publish the peer review history of their article (what does this mean?). If published, this will include your full peer review and any attached files.

Reviewer #1: No

Reviewer #2: No

---

## [Author Response · Author response to Decision Letter 1]

12 Jan 2020

dear Editor/reviewers, we, authors would like express our heart felt gratitude for giving us your comments and suggestions. we respond point by point for each raised concerns. so you can see the attached respones.

 thanks alot

---

## [Decision Letter · Decision Letter 2]

28 Jan 2020

Intention to donate blood and its predictors among adults of Gondar city: using theory of planned behavior

PONE-D-19-22717R2

Dear Dr. Kassie,

We are pleased to inform you that your manuscript has been judged scientifically suitable for publication and will be formally accepted for publication once it complies with all outstanding technical requirements.

With kind regards,

Sergio A. Useche, Ph.D.

Academic Editor

PLOS ONE

Additional Editor Comments (optional):

Reviewers' comments:

Reviewer's Responses to Questions

**Comments to the Author**

1. If the authors have adequately addressed your comments raised in a previous round of review and you feel that this manuscript is now acceptable for publication, you may indicate that here to bypass the “Comments to the Author” section, enter your conflict of interest statement in the “Confidential to Editor” section, and submit your "Accept" recommendation.

Reviewer #1: (No Response)

Reviewer #2: All comments have been addressed

2. Is the manuscript technically sound, and do the data support the conclusions?

Reviewer #1: No

Reviewer #2: (No Response)

3. Has the statistical analysis been performed appropriately and rigorously? 

Reviewer #1: Yes

Reviewer #2: (No Response)

4. Have the authors made all data underlying the findings in their manuscript fully available?

Reviewer #1: Yes

Reviewer #2: (No Response)

5. Is the manuscript presented in an intelligible fashion and written in standard English?

Reviewer #1: No

Reviewer #2: (No Response)

6. Review Comments to the Author

Reviewer #1: The authors have significantly improved the expression.

A few minor issues still arise:

“knowledge towards blood donation was assessed using ten questions which were previously used items in similar studies believed to be having similar demographic characteristics[16–18].”

Nearly 88.3% 212 were orthodox in religion and 80.58 were Amhara in ethnicity (Table 2). What type of orthodox?

Reviewer #2: (No Response)

7. PLOS authors have the option to publish the peer review history of their article (what does this mean?). If published, this will include your full peer review and any attached files.

Reviewer #1: No

Reviewer #2: Yes: Wilhellmuss Mauka

---

## [Editor Report · Acceptance letter]

5 Feb 2020

PONE-D-19-22717R2 

Intention to donate blood and its predictors among adults of Gondar city: using theory of planned behavior 

Dear Dr. Kassie:

I am pleased to inform you that your manuscript has been deemed suitable for publication in PLOS ONE. Congratulations! Your manuscript is now with our production department. 

With kind regards,

on behalf of

Dr. Sergio A. Useche 

Academic Editor

PLOS ONE